# Epidemiology and nomogram of pediatric and young adulthood osteosarcoma patients with synchronous lung metastasis: A SEER analysis

**Tao Liu[1], Lin Cui[2], Zongyun He[1], Zhe Chen[1], Haibing Tao[1], Jin Yang[1]\***

**1** Department of Hand and Foot Surgery, Yiwu Central Hospital, Yiwu, China, **2** Emergency Department, The 941st Hospital of the PLA Joint Logistic Support Force, Xining, China

\* yangjin546799@163.com

## Abstract

### Background

Patients with osteosarcoma and synchronous lung metastasis (SLM) have poor survival. This study aimed to explore the epidemiology data and construct a predictive nomogram to identify cases at risk of SLM occurrence among pediatric and young adulthood osteosarcoma patients.

### Methods

All data were extracted from Surveillance, Epidemiology, and End Results 17 registries. The age-standardized incidence rate (ASIR) and annual percentage change was evaluated, and reported for the overall population and by age, gender, race, and primary site. Univariate and multivariate logistic regression analyses were used to identify risk factors associated with SLM occurrence, then significant factors were used to develop the nomogram. The area under the receiver operating characteristic curve (AUC) and calibration curve were used to evaluated the predictive power of the nomogram. Survival analysis was assessed by the Kaplan-Meier method and the log-rank test. Multivariate Cox analysis was used to determine the prognostic factors.

### Results

A total of 278 out of 1965 patients (14.1%) presented with SLM at diagnosis. The ASIR increased significant from 0.46 to 0.66 per 1,000,000 person-years from year 2010 to 2019, with an annual percentage change of 3.5, mainly in patients with age 10–19 years, male and appendicular location. All patients were randomly assigned into train cohort and validation cohort with a spilt of 7:3. In the train cohort, higher tumor grade, bigger tumor size, positive lymph nodes and other site-specific metastases (SSM) were identified as significant risk factors associated with SLM occurrence. Then a nomogram was developed based on the four factors. The AUC and calibration curve in both train and validation cohorts demonstrated that the nomogram had moderate predictive power. The median cancer-specific survival

**Data Availability Statement:** All data are available from the Surveillance, Epidemiology, and End Results (SEER) Program website (www.seer.cancer.gov). Any registered researcher can

download this data for free (https://seer.cancer.gov/data). Others would be able to access or request these data in the same manner as the authors. The authors did not have any special access or request privileges that others would not have.

**Funding:** The author(s) received no specific funding for this work.

**Competing interests:** The authors have declared that no competing interests exist.

was 25 months. Patients with age 20–39 years, male, positive lymph nodes, other SSM were adverse prognostic factors, while surgery was protective factor.

## Conclusions

This study performed a comprehensive analysis regarding pediatric and young adulthood osteosarcoma patients had SLM. A visual, clinically operable, and easy-to-interpret nomogram model was developed for predicting the risk of SLM, which could be used in clinic and help clinicians make better decisions.

## Introduction

Osteosarcoma, the most common primary malignant bone tumor occurs in adolescence and those aged > 60 years [1]. This cancer affected male more frequently than females, with a ratio of 1.4:1 [2]. Nonetheless, it only represents less than 1% of all diagnosed cancer cases [3]. The incidence of osteosarcoma peaks at those with 10–19 years [3], and accounts for approximately 2% of children (1 to 14 years) and 3% of adolescents (15 to 19 years) with malignancy [4]. Different from occurring more frequently in axial locations among older patients, the osteosarcoma in young patients often arises in the metaphysis of long bones [5–8]. It might be associated with the rapid proliferation of bone, which is related to the growth spurt during puberty [2].

Compared to an improvement of long-term survival from 20% to over 70% among non-metastatic osteosarcoma patients, the survival of metastatic osteosarcoma patients remains poor, with only about 20% to 30% [9,10]. The poor survival of metastatic osteosarcoma is mainly due to the lung metastasis, which is the most prevalent metastatic type of metastatic osteosarcoma, accounting for more than 80% of the metastatic cases [11,12]. Thus, to identify patient at a high risk of lung metastasis and initiate timely treatment could be a way to improve the survival of these patients.

Currently, a few studies explored the risk factors of lung metastasis occurrence and constructed predictive models [10,13,14]. However, no study focuses on the risk factors for pediatric and young adulthood patients. Besides, the machine learning models are hard to interpret in clinical. Hence, this study aimed to describe the epidemiological trend of synchronous lung metastasis (SLM) at diagnosis in pediatric and young adulthood osteosarcoma patients, and identify risk factors and develop a clinically operable and easy-to-interpret nomogram model by using the big data from Surveillance, Epidemiology, and End Results (SEER) database.

## Methods

### SEER study population

The National Cancer Institute's (NCI's) SEER program collects the population-based cancer data in the US. In this part, all data were extracted from SEER 17 registries, November 2021 submission (2000–2019), representing approximately 26.5% of the US population. Because the details of site-specific metastasis, including bone, brain, liver, and lung metastases, are provided since year 2010, all data were extracted since then. The criteria used to identify eligible cases and the study design were presented in S1 Fig.

The following variables were extracted from the database, including year of diagnosis, age (1–9 years, 10–19 years, and 20–39 years), race (white, black, and others), gender (male and

female), primary site (appendicular [C40.0-C40.9] and axial [C41.0-C41.9]), tumor grade (low-grade [I/II], high-grade [III/IV], and unknown), tumor size (<5cm, 5-10cm, ≥10cm, unknown), lymph node status (negative, positive, and unknown), and site-specific metastasis. Except for synchronous lung metastasis, all other three site-specific metastases at diagnosis were grouped into one variable, i.e., SSM, with a yes denoted one or more other three site-specific metastases. Treatment information included surgery, radiotherapy, and systemic treatment.

### Age-standardized incidence rate

The age-standardized incidence rate (ASIR) was calculated by the SEER*Stat software (version 8.4.0.1; https://seer.cancer.gov/seerstat/). The incidence was age-standardized to the 2000 US standard population. The expression was presented as per 1,000,000 persons, and reported for the overall population and by age, gender, race, and primary site.

The annual percentage change (APC) of incidence was used to measure trends or the change in rates over time, which was quantified by the National Cancer Institute's (NCI's) Joinpoint Regression Program (version 4.9.0.0; https://surveillance.cancer.gov/joinpoint/). The APC was calculated by fitting a least squares regression line to the natural logarithm of the rates, using the calendar year as a regressor variable. This value was compared to zero to indicate statistics significance.

### Nomogram

All identified pediatric and young adulthood osteosarcoma patients were randomly divided into train dataset and validation dataset with a split of 7:3. The clinical characteristics comparison between train dataset and validation dataset was performed by Pearson's chi-square test. Univariate and multivariate logistic regression analyses were used to identify risk factors associated with SLM occurrence in the train dataset. Then a nomogram for predicting the probability of SLM occurrence at diagnosis was developed based on significant factors. The discriminative ability of the nomogram and each variable were evaluated by the area under the receiver operating characteristic curve (AUC). The calibration curve was used to measure the differences between the predicted and observed outcomes.

### Other statistical analyses

Baseline clinical characteristics of all pediatric and young adulthood osteosarcoma patients were described as frequencies and compared by Pearson's chi-square test. The cancer-specific survival (CSS) was assessed by the Kaplan-Meier method and the log-rank test. Multivariate Cox analysis was performed to determine the prognostic factors associated with CSS. The CSS was defined as the time from diagnosis of osteosarcoma to the death because of osteosarcoma. Cases died of other reasons or unknown reasons or alive were regarded as censored. The last follow-up time was 31 Dec, 2019. Two-sided *P-value < 0.05* was considered to indicate statistically significant difference.

## Results

### Population

A total of 1,965 pediatric and young adulthood osteosarcoma patients were identified. Among them, 278 out of 1965 patients presented with SLM at diagnosis, accounting for 14.1%. The comparison of baseline clinical characteristics between the two cohorts was summarized in Table 1. Compared to patients without SLM, those with SLM had high proportions of aging

**Table 1. Baseline characteristics comparison between pediatric and young adulthood osteosarcoma patients with and without synchronous lung metastasis.**

| Variables | Total, N = 1,965 (%) | Without SLM, N = 1,687 (%) | With SLM, N = 278 (%) | P-value |
|---|---|---|---|---|
| **Age (years)** | | | | < 0.001 |
| 1–9 | 203 (10) | 167 (10) | 36 (13) | |
| 10–19 | 938 (48) | 766 (45) | 172 (62) | |
| 20–39 | 824 (42) | 754 (45) | 70 (25) | |
| **Race** | | | | 0.418 |
| White | 1,486 (76) | 1,278 (76) | 208 (75) | |
| Black | 268 (14) | 224 (13) | 44 (16) | |
| Others | 211 (11) | 185 (11) | 26 (9) | |
| **Gender** | | | | 0.019 |
| Male | 1,099 (56) | 925 (55) | 174 (63) | |
| Female | 866 (44) | 762 (45) | 104 (37) | |
| **Year of diagnosis** | | | | 0.950 |
| 2010 | 152 (8) | 131 (8) | 21 (8) | |
| 2011 | 203 (10) | 179 (11) | 24 (9) | |
| 2012 | 188 (10) | 159 (9) | 29 (10) | |
| 2013 | 178 (9) | 153 (9) | 25 (9) | |
| 2014 | 221 (11) | 194 (11) | 27 (10) | |
| 2015 | 205 (10) | 176 (10) | 29 (10) | |
| 2016 | 203 (10) | 170 (10) | 33 (12) | |
| 2017 | 204 (10) | 177 (10) | 27 (10) | |
| 2018 | 207 (11) | 174 (10) | 33 (12) | |
| 2019 | 204 (10) | 174 (10) | 30 (11) | |
| **Primary site** | | | | 0.023 |
| Appendicular | 1,526 (78) | 1,295 (77) | 231 (83) | |
| Axial | 439 (22) | 392 (23) | 47 (17) | |
| **Tumor grade** | | | | < 0.001 |
| Low grade | 420 (21) | 412 (24) | 8 (3) | |
| High grade | 1,100 (56) | 898 (53) | 202 (73 | |
| Unknown | 445 (23) | 377 (22) | 68 (24) | |
| **Tumor size (cm)** | | | | < 0.001 |
| < 5 | 298 (15) | 288 (17) | 10 (4) | |
| 5–10 | 684 (35) | 619 (37) | 65 (23) | |
| ≥ 10 | 719 (37) | 559 (33) | 160 (58) | |
| Unknown | 264 (13) | 221 (13) | 43 (15) | |
| **Lymph node status** | | | | < 0.001 |
| Negative | 1,781 (91) | 1,548 (92) | 233 (84) | |
| Positive | 43 (2) | 20 (1) | 23 (8) | |
| Unknown | 141 (7) | 119 (7) | 22 (8) | |
| **Other SSM** | | | | < 0.001 |
| No | 1,892 (96) | 1,663 (99) | 229 (82) | |
| Yes | 73 (4) | 24 (1) | 49 (18) | |

SLM, synchronous lung metastasis; SSM, site-specific metastasis.

10–19 years (62 vs. 45%, *P < 0.001*), male patients (63 vs. 55%, *P < 0.019*), locating in appendicular (83 vs. 77%, *P = 0.023*), high grade (73 vs. 53%, *P < 0.001*), tumor size equal or larger than 10cm (58 vs. 33%, *P < 0.001*), positive lymph node (8 vs. 1%, *P < 0.001*) and other SSM (18 vs. 1%, *P < 0.001*).

### Trends in cancer incidence

The ASIR of all pediatric and young adulthood osteosarcoma patients changed from 3.31 per 1,000,000 person-years in year 2010 to 4.42 per 1,000,000 person-years in 2019, with no significant APC (1.7, 95%CI -0.7–4.1, *P = 0.141*). The ASIR of patients with SLM increased significant from 0.46 to 0.66 per 1,000,000 person-years from year 2010 to 2019, with an APC of 3.5 (95%CI, 0.9–6.1, *P = 0.015*) (Fig 1, S1 Table). Among patient with SLM, the ASIR was higher in subgroups of aging 10–19 years (ASIR 1.05–1.33 from 2010 to 2019; APC 4.1, *P = 0.098*), male patients (ASIR 0.59–0.74; APC 2.7, *P = 0.090*), and appendicular (ASIR 0.41–0.58; APC 4.1, *P = 0.020*) (Fig 2, Table 2 and S1 Table). The ASIRs among other subgroups showed no significant changes. Among different races, no obvious difference was found.

### Nomogram for predicting the probability of synchronous lung metastasis occurrence

All pediatric and young adulthood osteosarcoma patients were randomly assigned into train cohort and validation cohort with a spilt of 7:3. The comparison of clinical characteristics between these two cohorts was shown in S2 Table, which showed balanced clinical characteristics between two cohorts. Next, univariate and multivariate logistic regression analyses were performed with the data of train cohort. The results turned out that higher tumor grade, bigger tumor size, positive lymph nodes and other SSM were significant risk factors associated with SLM occurrence (S3 Table). And then a nomogram was developed for predicting the probability of synchronous lung metastasis occurrence with these four variables. As shown in Fig 3,

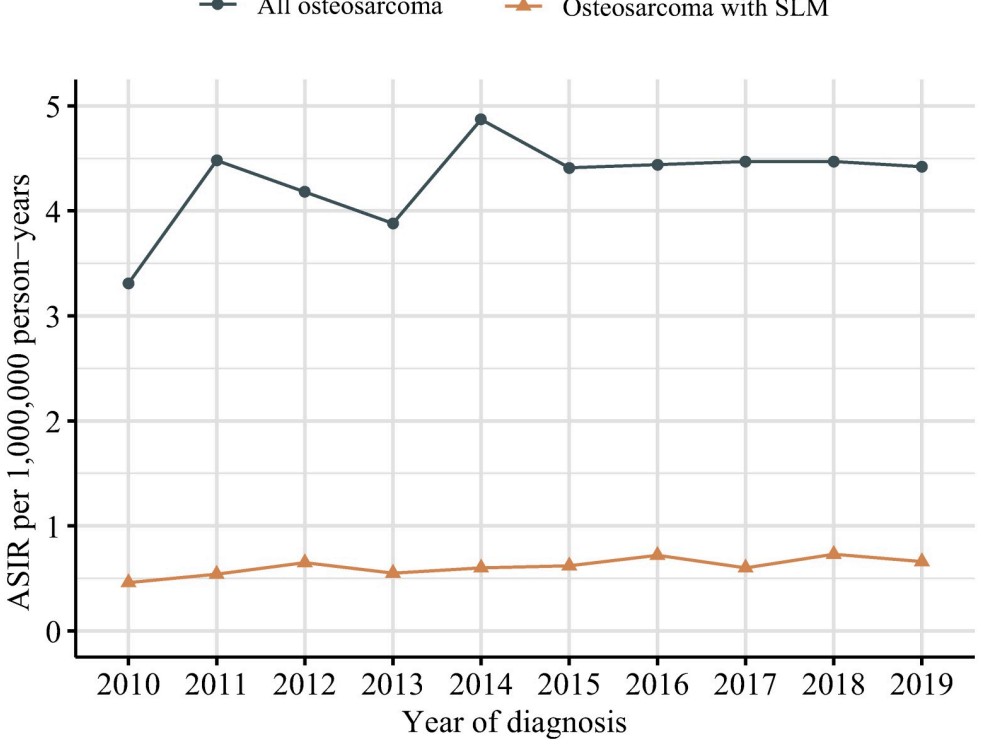

**Fig 1. Age-standardized incidence rate of pediatric and young adulthood osteosarcoma patient, United States, 2010 to 2019.** A) Overall, B) With synchronous lung metastasis. ASIR, age-standardized incidence rate; SLM, synchronous lung metastasis.

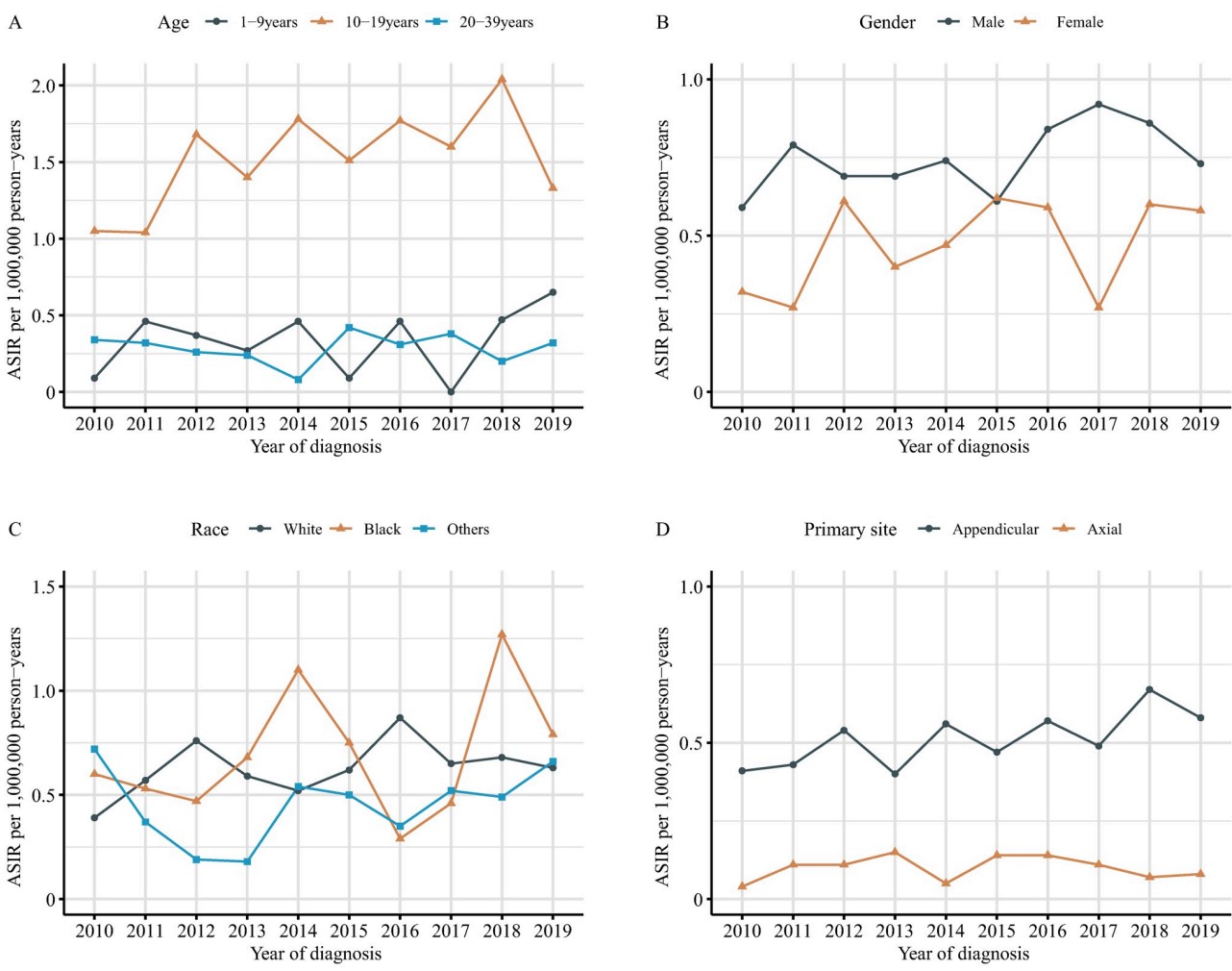

**Fig 2. Age-standardized incidence rate of pediatric and young adulthood osteosarcoma patient with synchronous lung metastasis, United States, 2010 to 2019.** A) By age, B) By gender, C) By race, D) By primary site. ASIR, age-standardized incidence rate.

each value of each variable was awarded a corresponding point according to the point scale, and the total points could be used for estimating the probability. The details of the points were summarized in S4 Table. According to the scale of total point, a total point less than 132 was regarded as low-risk group (a probability of risk less than 30%), a total point between 132 and 198 was regarded as median-risk group (a probability of risk between 30% and 70%), and a total point higher than 198 was regarded as high-risk group (a probability of risk over 70%).

To test the predictive ability of the nomograms, AUC and calibration plot were generated both in the train and validation cohorts. The AUC of the nomogram was 0.76 (95% CI 0.73–0.80) for train cohort and 0.76 (95% CI 0.71–0.81) for validation cohort, respectively. And that was much higher than each variable (Fig 4A and 4B). In addition, the calibration plots showed identified results between predicted probability and observed probability both in the train and validation cohorts (Fig 4C and 4D).

## Cancer-specific survival and prognostic factors

Among patients with SLM, approximately 96.4% and 74.8% of them respectively received systemic treatment and surgery during the first-course treatment. While approximately 12.2% of

**Table 2. Trends in age-standardized incidence rate of pediatric and young adulthood osteosarcoma patients with synchronous lung metastasis, United States, 2010 to 2019.**

|  | APC (95% CI) | *P* value |
|---|---|---|
| **Overall** | 3.5 (0.9–6.1) | *0.015* |
| **Age (years)** | | |
| 1–9 | 2.4 (-30.7–51.4) | *0.891* |
| 10–19 | 4.1 (-0.9–9.4) | *0.098* |
| 20–39 | 0.3 (-8.2–9.6) | *0.931* |
| **Race** | | |
| White | 2.9 (-2.3–8.5) | *0.239* |
| Black | 5.6 (-4.9–17.3) | *0.262* |
| Others | 1.5 (-7.6–11.4) | *0.729* |
| **Sex** | | |
| Male | 2.7 (-0.5–6.0) | *0.090* |
| Female | 4.8 (-3.0–13.2) | *0.204* |
| **Primary site** | | |
| Appendicular | 4.1 (0.8–7.4) | *0.020* |
| Axial | -0.4 (-11.1–11.5) | *0.933* |

APC, annual percent change; CI, confidence interval.

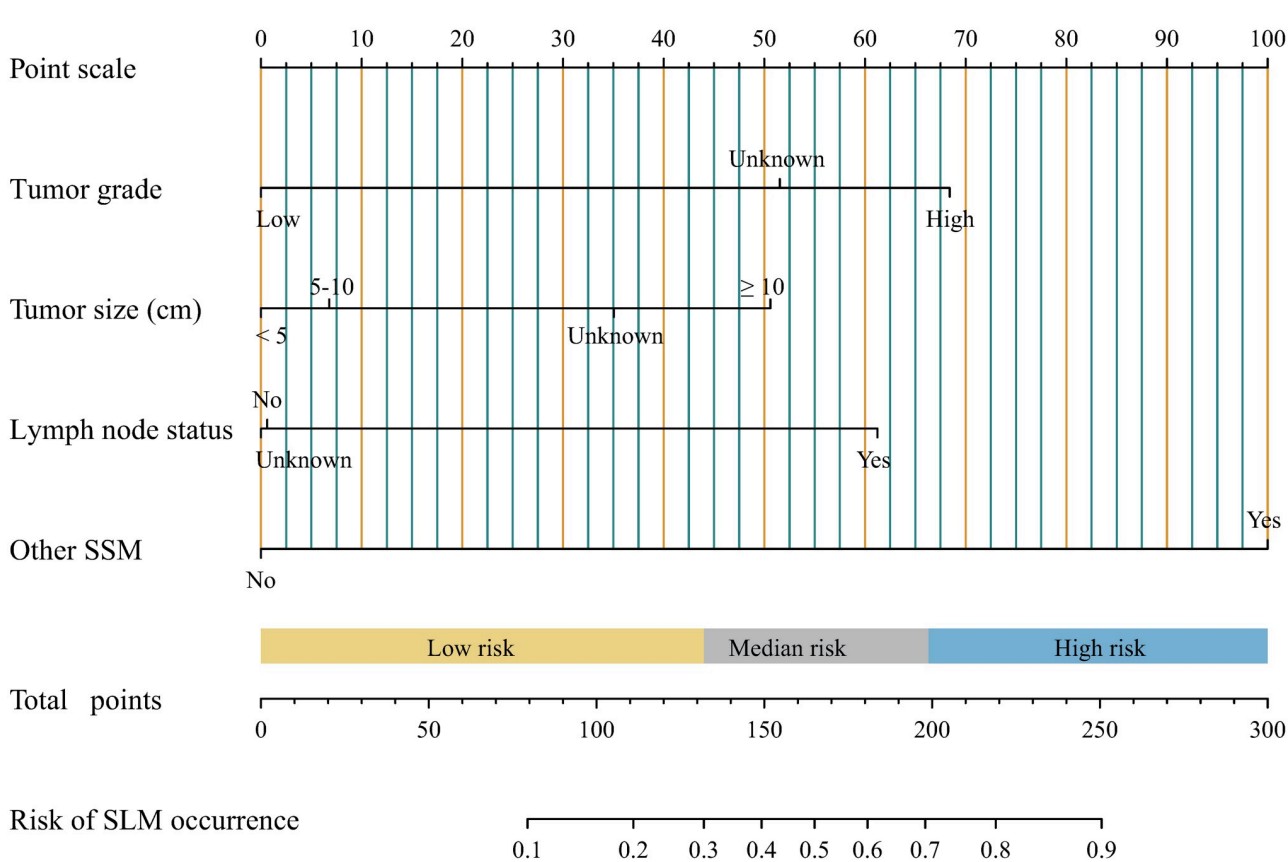

**Fig 3. A nomogram for predicting the risk of synchronous lung metastasis.**

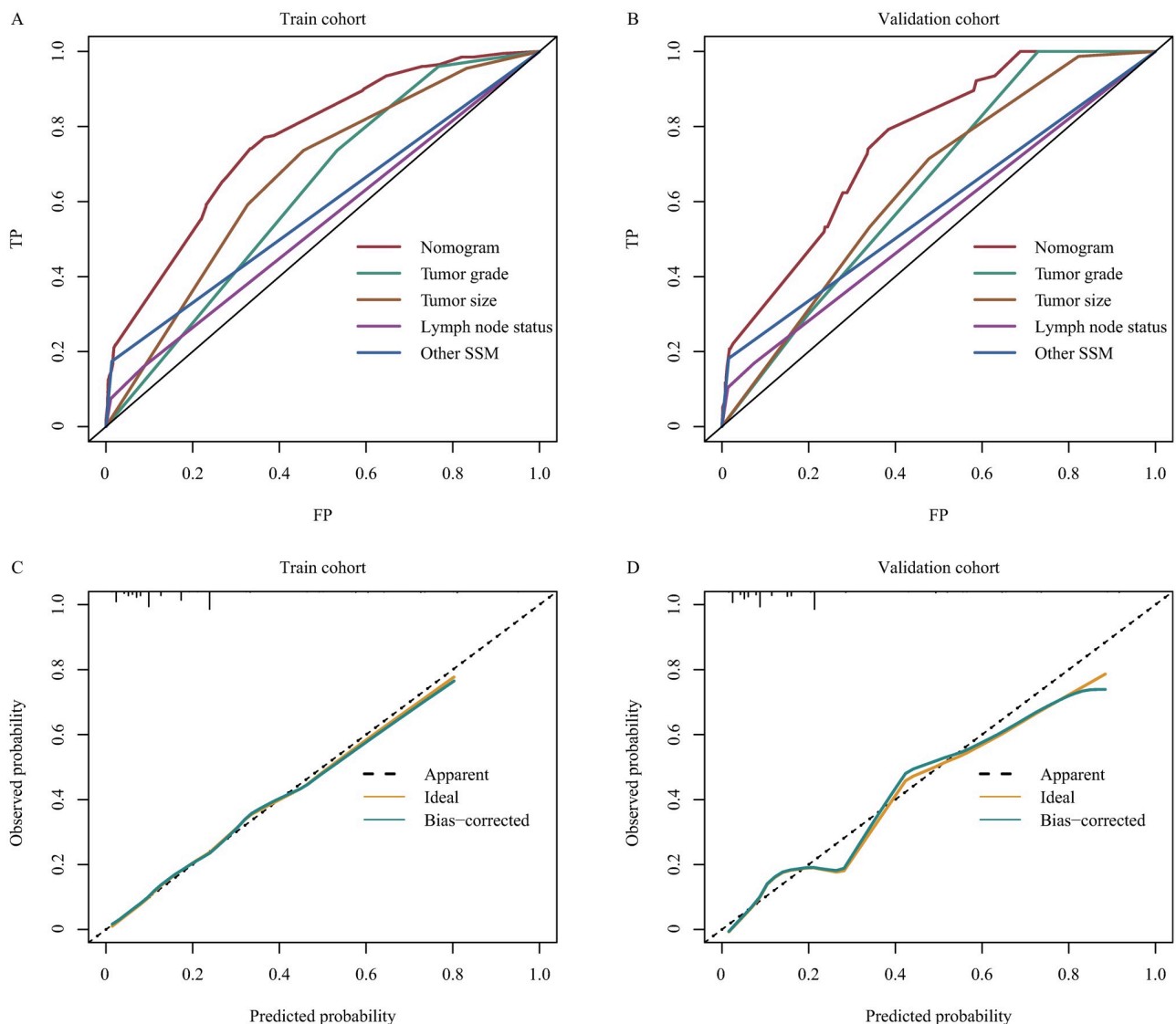

**Fig 4. Receiver operating characteristics curves and calibration curves for the nomogram.** A) Receiver operating characteristics curves in the train cohort, B) Receiver operating characteristics curves in the validation cohort, C) calibration curve in the train cohort, D) calibration curve in the validation cohort. SSM, site-specific metastasis.

them received radiotherapy. The median CSS was 25 months (95% CI 21–32 months, Fig 5). The multivariate Cox analysis turned out that patients with age 20–39 years, male, positive lymph nodes, other SSM had dismal survival (S5 Table). Surgery was a protective factor for CSS, while radiotherapy and systemic treatment had no obviously impact on CSS (S5 Table).

## Discussion

To the best of our knowledge, this is the first study of attempting to describe the epidemiology, and identify risk factors and prognostic factors for pediatric and young adulthood osteosarcoma patients with SLM, by using the SEER database. The result turned out that, between year 2010 and 2019, approximately 14.1% of all pediatric and young adulthood osteosarcoma patients developed a SLM at diagnosis. The rate was lower than that of entire osteosarcoma

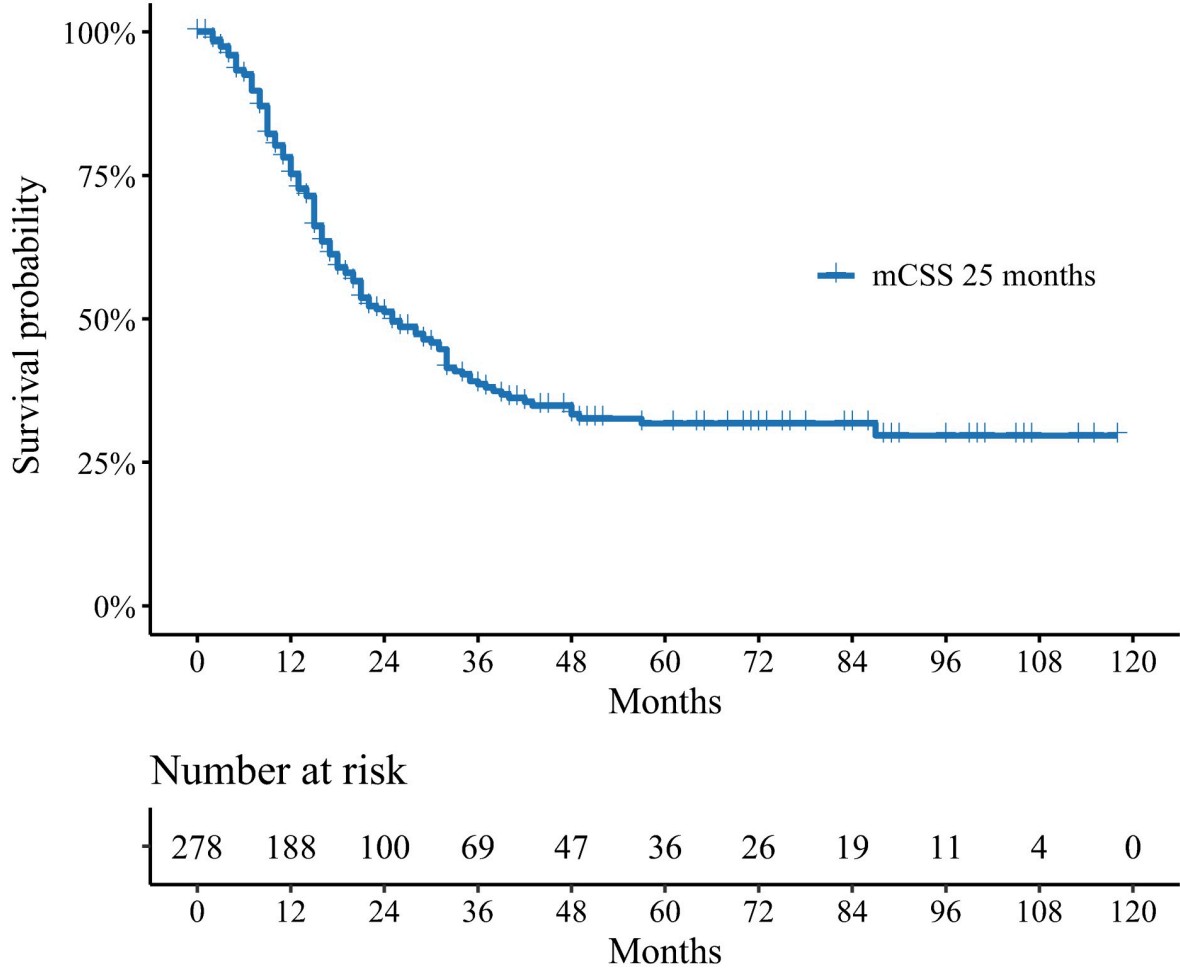

**Fig 5. Survival plot for pediatric and young adulthood osteosarcoma patient with synchronous lung metastasis.** CSS, cancer-specific survival.

patients with SLM, mainly because of a relatively higher proportion of SLM in the elder patients [13,14]. The incidence of pediatric and young adulthood osteosarcoma patients increased slightly with statistical significance from year 2010 to 2019. In addition, the increase was mainly pronounced among age 10–19 years, male patients, and appendicular location. However, no obvious difference was found among different races. Two earlier studies reported the epidemiology of all osteosarcoma patients, a higher incidence was also found in age 10–19 years, male patients, and appendicular location, respectively, comparted to the counterpart [1,3]. Besides a higher incidence was also found in black population and tumor grade of IV. But for difference tumor stage, the incidences were more pronounced in patients with localized and regional stages [1].

A few previous studies with SEER database reported clinical risk factors for lung metastases in entire osteosarcoma patients, including advanced age, gender, large tumor size, higher N stage, advanced tumor grade, presence of bone and/or brain metastases, and axial location [10,13–15]. In this study, among pediatric and young adulthood patients, we also found that advanced tumor grade, larger tumor size, positive lymph node, and other SSM were significant risk factors with SLM at diagnosis. Of note, gender and primary site were not associated with

SLM in this study. More importantly, we firstly constructed a visual, clinically operable, and easy-to-interpret nomogram model for predicting the risk of SLM in pediatric and young adulthood patients. The much higher AUC of nomogram, and the identified results between predicted outcome and observed outcome suggested that the nomogram had an obvious advantage in predicting the risk of SLM, which could be used in clinic and help clinicians make better decisions.

For nonmetastatic osteosarcoma patients, lung metastasis was also the most common site in the later stage of disease, with a rate of approximately 40–55% [16]. Besides, a study reported that approximately one out of three localized osteosarcoma patients would relapse, mainly due to the lung metastases [17]. The underlying mechanisms of lung metastasis is unclear, some potential factors might be associated with subsequent lung metastasis. A study demonstrated that cytoplasmatic monocyte ratio and neutrophil/lymphocyte ratio were associated with lung metastasis [18]. A meta-analysis found that clinical presentation with pathologic fractures could potentially increase the risk [19].

Nigris and colleagues demonstrated that the growth and development of osteosarcoma, as well as the metastasis development, depend on a tightly regulated and balanced angiogenesis [20]. The blocking of tumor angiogenesis could be a promising strategy for osteosarcoma therapy. In addition to that, multiple molecular regulation mechanisms may get involved in the metastasis of osteosarcoma. Studies demonstrated that MDM2 amplification [21], overexpression of VEGFR and PDGFR [22], the activation of mTOR [23], increase of IGF-1R [24], cyclin-dependent kinase [25], Aurora-B [26], TP53 [27] and MYC [28] were associated with the metastasis of osteosarcoma. And some inhibitors or potential inhibitors could suppress the metastasis of osteosarcoma, then extend the survival (S6 Table). In addition, some molecules may also be involved in metachronous lung metastases, such as HER-2 expression [29], cir-cFIRRE [30], CXCL1 [31], and extracellular vesicles miR-101 [32]. As lung metastasis is the most common involved organ at diagnosis, these factors could help for identifying SLM, and are worthwhile candidates for developing potential treatment regimen. Besides, Dong et.al demonstrated that a combination of eight genes, including RAB1, CLEC3B, FCGBP, RNASE3, MDL1, ALOX5AP, VMO1 and ALPK3, was associated with metastasis at diagnosis [33]. However, the molecular information are not provided in the SEER database. Further studies are warranted to include more molecular factors to construct a more powerful predictive model.

Some inflammatory cytokines may also get involved in the tumor progression, recurrence and metastasis of osteosarcoma [34]. A few studies demonstrated that the expression of IL-6, IL-8, and IL-1β could promote the tumor relapse and metastasis [35,36]. And drugs targeting IL-1 may be a potential method to improve the survival of metastatic osteosarcoma [37–40].

The most common and valuable method to screen lung metastasis is chest computed tomography (CT) [41]. But because of most of the lung metastasis in osteosarcoma patients are atypical, the sensitivity of CT scan widely ranges in the literature [42]. The improvement of new techniques may better help to identify lung metastasis and perform strictly chest surveillance, for example deep learning-based image reconstruction, 18F-fluorodeoxyglucose positron emission tomography/CT, volume doubling time and computer-aided diagnosis [42].

This study found a dismal CSS in pediatric and young adulthood osteosarcoma patients with SLM, with a median CSS of only 25 months, which was similar with previous studies [10,14]. The main prognostic factors for dismal survival consisted of age 20–39 years, male, positive lymph nodes, other SSM. Currently, the most common treatment for metastatic osteosarcoma is surgery plus chemotherapy, which could effectively improve the long-term survival of entire osteosarcoma patients [10]. In this study, among pediatric and young adulthood osteosarcoma patients, we only found an improvement of survival in surgical patients. But systemic treatment had no impact on survival, mainly because of most of patients (approximately

96.4%) received systemic treatment. Besides, no detail of the specific systemic treatment may also be a potential reason for the negative result. Cardiovascular side event was a common and fatal event, which was demonstrated to be the leading cause of non-cancer death among osteosarcoma patients [43]. For osteosarcoma patients, the most often reason for inducing cardiovascular death was chemotherapy [44]. Other studies suggested that radiation and targeted therapy could also induce cardiovascular toxicity and affect survival [45,46]. More importantly, a few studies reported that the osteosarcoma could directly invade the cardiovascular system and cause damage, with an incidence of approximately 2%, but it increased into 20% when performed autopsy [47–49].

Approximately 12.2% of patients received radiotherapy, but this treatment also demonstrated no impact on survival. A concern should be acknowledged, radiotherapy in young patients may cause second primary cancer as a late effect [50,51]. In addition to surgery and chemotherapy, vaccines, adoptive T-cell transfer, immune checkpoint inhibitors, and monoclonal antibodies may also ameliorate the survival outcome [52,53]. But more studies are warranted to better confirm the efficacy of these regimen, especially in combination with surgery.

There are some limitations in this study. First, the nature of retrospective study might lead to data bias. Second, though we select patients from the SEER database, the sample size of this study is still limited because of the rare incidence of osteosarcoma. Besides, we did not perform external validation. Hence, more samples are warranted to identify the predictive power of the nomogram. Third, the information available from SEER database are limited, many associated risk factors mentioned above are not provided. Further studies should include more factors to construct a more powerful model. Last, limited to the follow-up time, the long-term survival for pediatric and young adulthood osteosarcoma patients with SLM could not be estimated accurately. This should be investigated in the future studies.

## Conclusion

Approximately 14.1% of pediatric and young adulthood osteosarcoma patients had SLM at diagnosis. The incidence of this population increased slightly with statistical significance in last decade, mainly in patients with age 10–19 years, male and appendicular location. The survival of these patients was poor, and age, gender, positive lymph nodes, other SSM were associated prognostic factors. More importantly, we developed a visual, clinically operable, and easy-to-interpret nomogram model for predicting the risk of SLM, which could be used in clinic and help clinicians make better decisions.

## Supporting information

**S1 Checklist. STROBE statement—Checklist of items that should be included in reports of observational studies.**
(DOCX)

**S1 Fig. A flow chart of the study design.**
(TIF)

**S1 Table. Age-standardized incidence rate of pediatric and young adulthood osteosarcoma patients, United States, 2010 to 2019.** ASIR, age-standardized incidence rate; SEER, Surveillance, Epidemiology, and End Results. Rates are per 1,000,000 person-years.
(DOCX)

**S2 Table. Baseline characteristics comparison between train and validation cohorts after propensity score matching.** SSM, site-specific metastasis; SLM, synchronous lung

metastasis.
(DOCX)

**S3 Table. Risk factors associated with synchronous lung metastasis in pediatric and young adulthood osteosarcoma patients.** OR, odds ratio; CI, confidence interval; SSM, site-specific metastasis.
(DOCX)

**S4 Table. Nomogram point of each variable.** SSM, site-specific metastasis.
(DOCX)

**S5 Table. Prognostic factors associated with cancer-specific survival in pediatric and young adulthood osteosarcoma patients with synchronous lung metastasis.** HR, hazard ratio; CI, confidence interval; SSM, site-specific metastasis.
(DOCX)

**S6 Table. Summary of targeted molecular and related inhibitor associated with osteosarcoma metastasis.**
(DOCX)

## Acknowledgments

The authors are grateful to all the staff in the National Cancer Institute (USA) for their contribution to the SEER program.

## Author Contributions

**Conceptualization:** Tao Liu, Jin Yang.

**Data curation:** Tao Liu, Jin Yang.

**Formal analysis:** Tao Liu.

**Funding acquisition:** Jin Yang.

**Investigation:** Tao Liu, Lin Cui.

**Methodology:** Tao Liu, Lin Cui, Zhe Chen.

**Project administration:** Zongyun He, Zhe Chen, Haibing Tao, Jin Yang.

**Resources:** Tao Liu, Zongyun He, Haibing Tao.

**Software:** Lin Cui, Zongyun He, Haibing Tao.

**Supervision:** Lin Cui, Zongyun He, Haibing Tao, Jin Yang.

**Validation:** Zhe Chen, Haibing Tao, Jin Yang.

**Visualization:** Zhe Chen, Haibing Tao, Jin Yang.

**Writing – original draft:** Tao Liu, Zhe Chen, Haibing Tao, Jin Yang.

**Writing – review & editing:** Tao Liu, Jin Yang.

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
