## [Decision Letter · Decision Letter 0]

19 Jun 2023

PONE-D-23-15475Epidemiology and nomogram of pediatric and young adulthood osteosarcoma patients with synchronous lung metastasis: A SEER analysisPLOS ONE

Dear Dr. Yang,

Thank you for submitting your manuscript to PLOS ONE. After careful consideration, we feel that it has merit but does not fully meet PLOS ONE’s publication criteria as it currently stands. Therefore, we invite you to submit a revised version of the manuscript that addresses the points raised during the review process.

We look forward to receiving your revised manuscript.

Kind regards,

Filomena de Nigris, Ph.D.

Academic Editor

PLOS ONE

Journal Requirements:

4. Please include your tables as part of your main manuscript and remove the individual files. Please note that supplementary tables (should remain/ be uploaded) as separate "supporting information" files

Reviewers' comments:

Reviewer's Responses to Questions

**Comments to the Author**

1. Is the manuscript technically sound, and do the data support the conclusions?

Reviewer #1: Yes

Reviewer #2: Yes

2. Has the statistical analysis been performed appropriately and rigorously? 

Reviewer #1: Yes

Reviewer #2: Yes

3. Have the authors made all data underlying the findings in their manuscript fully available?

Reviewer #1: Yes

Reviewer #2: Yes

4. Is the manuscript presented in an intelligible fashion and written in standard English?

Reviewer #1: Yes

Reviewer #2: Yes

5. Review Comments to the Author

Reviewer #1: Minor revision

1. Since the growth and development of osteosarcoma depend on a tightly regulated and balanced angiogenesis and the blocking of tumor angiogenesis could be a promising strategy for cancer therapy, it is necessary to report in the Introduction some targets, which have already been identified.

An example is represented by cyclin-dependent kinases, fundamental mediators of neoangiogenesis and other (see de Nigris F, et al. Osteosarcoma cells induces endothelial cell proliferation during neo-angiogenesis. J Cell Physiol. 2013 Apr;228(4):846-52).

2. To organize a descriptive table of molecular targets, which are altered in the presence of osteosarcoma metastases.

Reviewer #2: Manuscript titled "Epidemiology and nomogram of pediatric and young adulthood osteosarcoma patients with synchronous lung metastasis: A SEER analysis" is a very interesting analysis of SLM risk in patients with osteosarcoma. The overall structure of the manuscript is of good quality, figures are clear and easy to understand. Methods and results are of good quality and support the initial hypothesis. Authors should improve the manuscript in some parts:

1. authors should add information on the pathophysiology of SLM events in these patients and the involvement of serum cytokines like IL-1, IL-6 and CXCL-12 in cancer metastasis as well as in cardiovascular side events.

2. In discussion, authors should add some data on the incidence of tumour-related and chemotherapy-related cardiovascular side events in these patients ( long term toxicity).

Manuscript will be accepted after minor revision.

6. PLOS authors have the option to publish the peer review history of their article (what does this mean?). If published, this will include your full peer review and any attached files.

Reviewer #1: No

Reviewer #2: No

---

## [Author Response · Author response to Decision Letter 0]

26 Jun 2023

Dear Editor,

Thank you very much for your e-mail on 20 JUN 2023 to informing us to revise our manuscript (PONE-D-23-15475) and respond to the reviewer(s)' comments.

Here, we have carefully studied the reviewers’ comments and made the appropriate revisions to our manuscript.

The following are our specific responses to the editor’s and reviewers’ comments:

Author response: Thanks for the editor’s comment. We have amended the style according to the requirements.

2. We note that you have indicated that data from this study are available upon request. PLOS only allows data to be available upon request if there are legal or ethical restrictions on sharing data publicly. For more information on unacceptable data access restrictions, please see http://journals.plos.org/plosone/s/data-availability#loc-unacceptable-data-access-restrictions. In your revised cover letter, please address the following prompts:

Author response: Thanks for the editor’s comment. All the data used in this study were obtained form the public database, SEER database, which is open for all researchers. We have provided sufficient patients selection criteria. Hence, we think there is no need to upload the minimal anonymized data set to replicate our study findings. Because they could replicate this study by access the SEER database for original data, which is more convincing. If needed, we could provide all the datasets used in this study upon reasonable request, by inquiring the corresponding author.

Author response: Thanks for the editor’s comment. Revised as required.

4. Please include your tables as part of your main manuscript and remove the individual files. Please note that supplementary tables (should remain/ be uploaded) as separate "supporting information" files

Author response: Thanks for the editor’s comment. Revised as required.

Author response: Thanks for the editor’s comment. Revised as required.

Author response: Thanks for the editor’s comment. We have added some references in the discussion section. We have reviewed our reference list and ensure that it is complete and correct.

Reviewer #1: Minor revision

1. Since the growth and development of osteosarcoma depend on a tightly regulated and balanced angiogenesis and the blocking of tumor angiogenesis could be a promising strategy for cancer therapy, it is necessary to report in the Introduction some targets, which have already been identified.

An example is represented by cyclin-dependent kinases, fundamental mediators of neoangiogenesis and other (see de Nigris F, et al. Osteosarcoma cells induces endothelial cell proliferation during neo-angiogenesis. J Cell Physiol. 2013 Apr;228(4):846-52).

Author response: Thanks for the review’s comment. We have added this reference (ref. 20) in the Discussion section, paragraph 4, “Nigris and colleagues demonstrated that the growth and development of osteosarcoma, as well as the metastasis development, depend on a tightly regulated and balanced angiogenesis[20]. The blocking of tumor angiogenesis could be a promising strategy for osteosarcoma therapy.”

2. To organize a descriptive table of molecular targets, which are altered in the presence of osteosarcoma metastases.

Author response: Thanks for the review’s comment. We have added some discussion about the potential molecular targets in the presence of osteosarcoma metastases. And summarize them in eTable 6. See details in paragraph 4 of the Discussion, “In addition to that, multiple molecular regulation mechanisms may get involved in the metastasis of osteosarcoma. Studies demonstrated that MDM2 amplification[21], overexpression of VEGFR and PDGFR[22], the activation of mTOR[23], increase of IGF-1R[24], cyclin-dependent kinase[25], Aurora-B[26], TP53[27] and MYC[28] were associated with the metastasis of osteosarcoma. And some inhibitors or potential inhibitors could suppress the metastasis of osteosarcoma, then extend the survival (eTable 6). In addition, some molecules may also be involved in metachronous lung metastases, such as HER-2 expression[29], circFIRRE[30], CXCL1[31], and extracellular vesicles miR-101[32]. As lung metastasis is the most common involved organ at diagnosis, these factors could help for identifying SLM, and are worthwhile candidates for developing potential treatment regimen.”

Reviewer #2: Manuscript titled "Epidemiology and nomogram of pediatric and young adulthood osteosarcoma patients with synchronous lung metastasis: A SEER analysis" is a very interesting analysis of SLM risk in patients with osteosarcoma. The overall structure of the manuscript is of good quality, figures are clear and easy to understand. Methods and results are of good quality and support the initial hypothesis. Authors should improve the manuscript in some parts:

1. authors should add information on the pathophysiology of SLM events in these patients and the involvement of serum cytokines like IL-1, IL-6 and CXCL-12 in cancer metastasis as well as in cardiovascular side events.

Author response: Thanks for the review’s comment. We add some discussion about the involved cytokines in the paragraph 5 of Discussion section, “Some inflammatory cytokines may also get involved in the tumor progression, recurrence and metastasis of osteosarcoma[34]. A few studies demonstrated that the expression of IL-6, IL-8, and IL-1β could promote the tumor relapse and metastasis[35,36]. And drugs targeting IL-1 may be a potential method to improve the survival of metastatic osteosarcoma[37-40].”

2. In discussion, authors should add some data on the incidence of tumour-related and chemotherapy-related cardiovascular side events in these patients ( long term toxicity).

Author response: Thanks for the review’s comment. We add some discussion about the cardiovascular side events in the paragraph 7 of Discussion section, “Cardiovascular side event was a common and fatal event, which was demonstrated to be the leading cause of non-cancer death among osteosarcoma patients[43]. For osteosarcoma patients, the most often reason for inducing cardiovascular death was chemotherapy[44]. Other studies suggested that radiation and targeted therapy could also induce cardiovascular toxicity and affect survival[45,46]. More importantly, a few studies reported that the osteosarcoma could directly invade the cardiovascular system and cause damage, with an incidence of approximately 2%, but it increased into 20% when performed autopsy[47-49]”.

We hope that we have adequately addressed the constructive comments. We greatly appreciate your consideration of this manuscript for possible publication in your journal and look forward to hearing from you soon.

Sincerely,

Dr. Jin Yang

---

## [Editor Report · Decision Letter 1]

29 Jun 2023

Epidemiology and nomogram of pediatric and young adulthood osteosarcoma patients with synchronous lung metastasis: A SEER analysis

PONE-D-23-15475R1

Dear Dr. Yang

We’re pleased to inform you that your manuscript has been judged scientifically suitable for publication and will be formally accepted for publication once it meets all outstanding technical requirements.

Kind regards,

Filomena de Nigris, Ph.D.

Academic Editor

PLOS ONE
---

## [Editor Report · Acceptance letter]

4 Jul 2023

PONE-D-23-15475R1 

Epidemiology and nomogram of pediatric and young adulthood osteosarcoma patients with synchronous lung metastasis: A SEER analysis 

Dear Dr. Yang:

I'm pleased to inform you that your manuscript has been deemed suitable for publication in PLOS ONE. Congratulations! Your manuscript is now with our production department. 

Kind regards, 

on behalf of

Prof. Filomena de Nigris 

Academic Editor

PLOS ONE